# Osteocyte Dysfunction in Joint Homeostasis and Osteoarthritis

**DOI:** 10.3390/ijms22126522

**Published:** 2021-06-17

**Authors:** Lanlan Zhang, Chunyi Wen

**Affiliations:** 1Department of Biomedical Engineering, Faculty of Engineering, The Hong Kong Polytechnic University, Hong Kong 999077, China; 18080922d@connect.polyu.hk; 2Research Institute for Smart Ageing, The Hong Kong Polytechnic University, Hong Kong 999077, China

**Keywords:** osteoarthritis, osteocyte, bone remodeling, articular cartilage, sclerostin

## Abstract

Structural disturbances of the subchondral bone are a hallmark of osteoarthritis (OA), including sclerotic changes, cystic lesions, and osteophyte formation. Osteocytes act as mechanosensory units for the micro-cracks in response to mechanical loading. Once stimulated, osteocytes initiate the reparative process by recruiting bone-resorbing cells and bone-forming cells to maintain bone homeostasis. Osteocyte-expressed sclerostin is known as a negative regulator of bone formation through Wnt signaling and the RANKL pathway. In this review, we will summarize current understandings of osteocytes at the crossroad of allometry and mechanobiology to exploit the relationship between osteocyte morphology and function in the context of joint aging and osteoarthritis. We also aimed to summarize the osteocyte dysfunction and its link with structural and functional disturbances of the osteoarthritic subchondral bone at the molecular level. Compared with normal bones, the osteoarthritic subchondral bone is characterized by a higher bone volume fraction, a larger trabecular bone number in the load-bearing region, and an increase in thickness of pre-existing trabeculae. This may relate to the aberrant expressions of sclerostin, periostin, dentin matrix protein 1, matrix extracellular phosphoglycoprotein, insulin-like growth factor 1, and transforming growth factor-beta, among others. The number of osteocyte lacunae embedded in OA bone is also significantly higher, yet the volume of individual lacuna is relatively smaller, which could suggest abnormal metabolism in association with allometry. The remarkably lower percentage of sclerostin-positive osteocytes, together with clustering of Runx-2 positive pre-osteoblasts, may suggest altered regulation of osteoblast differentiation and osteoblast-osteocyte transformation affected by both signaling molecules and the extracellular matrix. Aberrant osteocyte morphology and function, along with anomalies in molecular signaling mechanisms, might explain in part, if not all, the pre-osteoblast clustering and the uncoupled bone remodeling in OA subchondral bone.

## 1. Introduction

Osteoarthritis (OA) is the most common form of arthritic disease, mainly afflicting the load-bearing joints, e.g., knee and hip. It is also recognized as a major cause of joint pain and disability, contributing to a compromised quality of life in older adults [1,2]. Articular cartilage, subchondral bone, and synovium are among the tissues that may display abnormalities in OA. The radiological features of osteoarthritic subchondral bone include sclerosis, cyst and osteophyte formation on plain x-radiograph, and bone marrow lesions (BMLs) on magnetic resonance imaging, all of which have been proven to underline the anomalies of bone mineralization [3,4]. Such impaired mineralization was noted from all parts of the trabecular bone, along with compositional changes of the bone matrix, e.g., a low ratio of mineral/collagen content [5,6]. Sclerotic changes and osteophyte formation are both believed to arise from elevated bone turnover with an increase in osteoblastic over osteoclastic activities [7,8]. BMLs and sclerosis are likely associated with abnormal signaling related to sclerostin, periostin, and dentin matrix protein 1 (DMP-1) [9,10,11]. Meanwhile, cysts surrounded by less mineralized bone and osteoid formation uncoupled with mineralization may suggest spatially differential regulation of osteoblasts and osteoclasts by Wnt/β-catenin signaling and the OPG/RANKL/RANK pathway [12,13]. Modulation of bone remodeling was able to reduce osteophyte formation and alleviate cartilage degeneration in experimental models of OA and helped with restricting BMLs in human OA [14,15,16]. However, the pathomechanism of osteoarthritic bone remodeling and the resultant structural disturbances has not been fully understood yet.

Bone resorption and formation are tightly controlled and precisely coordinated by bone cells during the bone remodeling process to maintain the homeostasis of adult bone metabolism [17]. The structural and functional disturbances of the osteoarthritic bone are generally attributed to primary osteoblastic dysfunction [18]. Osteoarthritic osteoblasts showed blunted responses to the stimulus of cytokines [19,20] and an aberrant ratio of mineral and collagen production [6]. Meanwhile, osteocyte dysfunction and abnormal osteoblast-osteocyte differentiation are also recognized as the possible causes of OA [21]. Osteocytes compose 90–95% of bone cells and are responsible for mechano-transduction. It is known that osteocytes may respond to mechanical stimuli like micro-cracks with death. Following the deaths of osteocytes, bone remodeling is initiated via activation of osteoclastogenesis and bone resorption [17]. Osteocytes also act as the key regulators of osteoblast differentiation through the secretion of sclerostin. Through suppressing Wnt signaling, sclerostin can inhibit osteoblastic activity and terminate the cycle of bone remodeling [17,22,23]. There is growing evidence suggesting that osteocytes act as a central regulator of bone remodeling to maintain bone homeostasis and integrity [24]. However, the exact pathophysiology of osteocytes and their contributions to the structural and functional disturbances of the osteoarthritic bone remain unelucidated. Hereby, this review aimed to discuss the morphology and function of osteocytes and their association with subchondral bone remodeling in joint homeostasis and OA, along with the molecular pathways potentially involved in pathogenesis.

## 2. Disturbances of Subchondral Bone

The role of subchondral bone in OA progression has been well documented. Disturbances of the subchondral bone are characterized by anomalous bone morphology at the tissue level, irregular osteocyte activities at the cellular level, and altered protein expressions at the molecular level. The Wnt/β-catenin pathway, which is responsible for osteogenesis and bone remodeling, is enriched in osteoarthritic subchondral bones [12]. Sclerostin, a major Wnt inhibitor, is expressed in calcified cartilage and subchondral bone. Its deficiency is often associated with OA development, likely through activating Wnt signaling [25]. In both OA and sclerostin-deficient models, an increase in bone volume fraction and trabecular bone thickness were observed in the subchondral bone, along with aberrant distribution of bone mineral density (BMD) [25,26,27,28]. This change is believed to be load-dependent, as higher bone mass is usually found excessively concentrated in the load-bearing region, with relatively lower bone mass in the less or non-loading region. For instance, our data suggested that osteoarthritic bone exhibited unevenly distributed bone mass from the load-bearing to non-load-bearing regions, such as in the superior portion of femoral heads underneath the articular cartilage. In contrast, there was no significant difference in the bone mass organization of the femoral heads from the control group (Figure 1a,b). In accordance with structural changes, sclerostin levels are also believed to be modulated by loading. Expression of sclerostin by osteocytes was downregulated by mechanical loading while upregulated by unloading in mice, suggesting a role in mechano-transduction [29,30]. Overexpression of SOST, the gene encoding sclerostin, was found to inactivate Wnt signaling, attenuate bone formation, and reduce mineralization in response to mechanical stimulation (Figure 1c). Suppressed bone remodeling then resulted in little to no difference in load-bearing and non-loading regions [30]. Hence, changes in sclerostin levels likely underlie the occurrence of sclerotic lesions preferentially in load-bearing regions. It might thus be considered as an adaption to support an abnormal level of compressive stress, such as when cartilage tissues are severely eroded [26,31] or when repetitive loading is applied [32]. 

Subchondral trabeculae are generally well-aligned with the direction of principal compressive stress, suggesting adaptation to loading conditions [26,27]. Periostin, a matricellular protein produced by osteocytes and osteoblasts, may participate in the regulation of bone formation and resorption associated with sclerostin. Increased periostin expression was observed upon mechanical stimulation, which was followed by a decreased sclerostin expression (Figure 1c) [10,33]. Periostin expression was enhanced in OA in subchondral bone, synovial fluid, and cartilage [34,35,36]. It was thought to exacerbate OA by promoting the expressions of pro-inflammatory cytokines, such as interleukin (IL)-6 and -8, and matrix metalloproteinase (MMP) production, such as through nuclear factor kappa B (NFκB) signaling [35]. High periostin level was linked with decreased bone surface/bone volume ratio, increased trabecular number, low structural model index (SMI), and high stiffness [10,11]. Such changes echoed the findings in OA, specifically the drop in bone surface area, the elevated number of trabeculae bones and their plate-like microarchitecture, as well as the stiffening of subchondral bones [26,27]. Plate-like trabecular bones were positively correlated with OA severity [37] and were generally able to sustain higher mechanical stress [38]. Meanwhile, stiffening of subchondral bone could cause ineffective load transfer from cartilage to bone. This was proposed as a possible source of cartilage damage and might explain the exacerbation of OA under excessive loading [39]. Thus, periostin might also induce cartilage degeneration by disturbing the underlying subchondral bone through sclerostin-Wnt signaling [11,40]. Interestingly, other characteristics of osteoarthritic bones, such as decreased trabeculae connectivity and BMD, were identified in periostin-deficient models [10,11]. Likewise, inhibition of sclerosing activities through antibodies resulted in higher connectivity density and BMD, which appeared to contradict the findings in OA [41,42]. Nevertheless, it should be noted that hyper-mineralization of the subchondral bone was also observed in early-stage OA [43]. Therefore, periostin and sclerostin might first affect bone mineralization in early-stage OA, while impaired mineralization in end-stage OA could relate to cellular dysfunction. Other signaling pathways might also be responsible for these sclerotic changes [44], such as transforming growth factor-beta (TGF-β) signaling [45], whose inhibition increased connectivity density and attenuated OA, DMP-1, whose deletion in cartilage led to an osteoarthritic phenotype [9,46].

Subchondral bone marrow lesions (BMLs) have been characterized by sclerotic bones with less mineralization [3] and are believed to predict the progression of OA [47,48,49,50]. Our data also confirmed that the inorganic content of bone plugs from knee OA specimens was significantly lower (Figure 2a,b). We have also observed osteoid formation near trabeculae or in the marrow space of OA bone (Figure 3g,i), which were compatible with that of BMLs. Aberrant mineralization is likely associated with DMP-1. DMP-1 is responsible for the mineralization of collagen content, as it is critical for mineral nucleation [51]. Decreased DMP-1 expression and disrupted mineralization of the extracellular matrix (ECM) is accompanied by the activation of Wnt/β-catenin signaling, along with sclerostin deficiency [52]. Altered DMP-1 signaling might explain why OA bone exhibited higher mass yet with decreased structural connectivity and reduced mineral density. However, DMP-1 expression by osteocytes was higher in human OA samples [28]. Meanwhile, mechanical loading appeared to upregulate DMP-1 expression by osteocytes in both bone formation and resorption sites, while osteoblasts expressed less DMP-1 in the early stage of induction [53]. Hence, decreased DMP-1 expression in osteoblasts was proposed as a possible cause of low BMD in OA. This was supported by the results of our research, which showed the formation of osteoid-like tissues in marrow space together with the clustering of bone cells in OA (Figure 3i). Low BMD and poor connectivity were associated with decreased strength. Despite being plate-like and well-aligned, osteoarthritic bones were also weaker than normal bones with similar morphology [54,55]. Therefore, although the initial attempt was to increase BMD in response to mechanical loading, bone remodeling in OA might eventually render the subchondral bone mechanically inferior rather than superior.

### Crosstalk with Cartilage Degeneration

Disturbances of the subchondral bone have been closely associated with cartilage degeneration. For example, the size of the tibial subchondral bone might predict the overlying articular cartilage defect and OA severity [56]. Insulin-like growth factor 1 (Igf1) deficiency was linked with decreased bone formation, increased osteoid formation, and delayed and diminished mineral deposition in the subchondral bone (Figure 4) [57]. In OA, Igf1 expression by osteoblasts was upregulated [19,36]. Similarly, the mechanical strain was believed to increase Igf1 levels in osteocytes [58,59]. Hence, Igf1 was thought to facilitate the drop in sclerostin level upon mechanical loading and thus modulate bone remodeling [60]. In contrast, Igf1 expression by osteoarthritic chondrocytes was reduced [61]. This might be attributed to the presence of stressors, such as reactive oxygen species (ROS) [62], a well-identified cause of cartilage damage [63]. The addition of Igf1 to articular cartilage could increase matrix biosynthesis or enhance the stimulating effects of dynamic mechanical loading [64]. Likewise, matrix degradation in response to static loading could also be rescued by Igf1, which might suggest a role of Igf1 in maintaining the balance of anabolism and catabolism [65]. Cartilage degeneration is thus likely linked with both decreased anabolic activities partly due to Igf1 deficiency and increased catabolism associated with elevated expressions of degrative enzymes, such as MMPs [28]. 

Bone remodeling mediated by Wnt signaling is believed to stiffen trabecular bones and alter their micro-architecture. For instance, in spontaneous OA models of guinea pigs, changes from rod-like to plate-like morphology were thought to precede or occur simultaneously with cartilage degeneration [66,67]. Past experiments also observed subchondral bone remodeling beneath intact cartilage in human OA, but only significant when cartilage tissues were damaged [26,66]. Ineffective load transfer due to structural disturbances of the subchondral bone might increase the strain endured by cartilage. High mechanical strain could then increase ROS production by chondrocytes, presumably because of mitochondrial deformation (Figure 4) [68,69]. Therefore, differential expressions of Igf1 in bone and cartilage, for instance, could be involved in the progression of OA. Similarly, while TGF-β could promote chondrocyte proliferation and exhibited protective effects in articular cartilage, its elevation in the subchondral bone was associated with cartilage loss and OA development [45]. The crosstalk between subchondral bone remodeling and cartilage degeneration was also believed to be regulated by MMP-13, which was less expressed by osteoarthritic osteocytes. Insufficient expression of MMP-13 was assumed to increase trabecular bone volume and cause cartilage degeneration independent of other stimuli. A reduction in osteocyte-derived MMP-13 was associated with disrupted lacuno-canalicular networks (LCN), which likely affected mechano-transduction and bone remodeling in early-stage OA. Meanwhile, expressions of proteins responsible for metabolism or matrix degradation, such as MMP-13 expression by chondrocytes, were also induced by insufficient osteocytic MMP-13 [43]. Hence, bone remodeling might start before cartilage degeneration and contribute to the aberrant metabolism in cartilage. With accumulated micro-damage, such as when an excessive load was applied or when cartilage was lost [66], mechanically inferior bones would be formed predominantly, and OA exacerbated. The above findings suggested that osteocytes could regulate trabecular bone morphology, bone mineralization, and ultimately bone and cartilage homeostasis in response to mechanical loading. Hence, the interplay between cartilage and subchondral bone is worth further investigation and should be extended to cellular and protein levels.

## 3. Osteocyte Dysfunction

Bone remodeling is a complex but precisely coordinated biological process involving osteoblasts, osteoclasts, and mesenchymal stem cells (MSCs), all of which are likely mediated by osteocytes [17]. Osteoclasts dissolve the minerals and collagens of the bone matrix; MSCs are recruited to bone formation sites and then differentiate into osteoblasts for collagen deposition and mineralization. Osteocytes are the major source of sclerostin, DMP-1, and matrix extracellular phosphoglycoprotein (MEPE) in bone [70], which designates its role in osteogenesis, mechano-transduction [22,71,72], and bone mineralization [9,73,74]. The mechano-sensitivity of osteocytes is facilitated by integrins, cytoskeleton, stretch-activated ion channels, gap junctions, and the LCN [21,75]. Integrins and focal adhesions are essential for mediating the response of gap junctions to mechanical stimulation and for the activation of Wnt signaling and other pathways in osteocytes. It has been proposed that mechanical deformations can induce ionic fluxes in LCN. The ionic currents can both amplify tissue strains and produce shear stress to activate osteocytes [13,21]. Osteocytes can sense the strain on the cytoskeleton. Upon stimulation, they will secrete signaling molecules to regulate osteoclastic and osteoblastic activities and modulate the recruitment of mesenchymal progenitors, which eventually results in bone remodeling. Disruption of microtubules resulted in a blunted decrease in sclerostin expression upon mechanical loading [76]. Similarly, Kindlin-2, a protein responsible for cytoskeleton organization and the formation of focal adhesions, was essential to the suppression of sclerostin expression through inhibiting Smad2/3 signaling [77]. These findings again confirm the role of the cytoskeleton in mechanical sensing. In OA, osteocyte dysfunction is characterized by altered lacuna morphology and protein expressions, both of which may be the response to excessive mechanical loading. The morphological and functional aberrations then cause dysregulation of bone cells, disruption of bone and cartilage homeostasis, and OA exacerbation [21,78,79]. Hence, the cellular and molecular mechanisms behind such anomalies should be elucidated.

### 3.1. Osteocyte Morphology

Osteocytes can actively change their microenvironment, notably the ECM and the lacunae where they reside. An altered extracellular environment can facilitate differential responses to mechanical stress. The morphology and alignment of osteocyte lacunae are assumed to be correlated with mechano-transduction. Notably, changes in the LCN network may affect the sensitivity of individual osteocytes to matrix deformation [21]. A decrease in LCN area seen in early-stage OA, for instance, might suggest that MMP-13 deficiency in osteocytes contributed to the reduced mechano-sensitivity of lacunae (Figure 5a) [43]. The uCT images obtained by the authors’ team revealed altered morphology of osteocyte lacunae in OA, where their three-dimensional structure appeared more plate-like rather than rod-like (Figure 3e). Plate-like architecture indicated fragility and reduced sensitivity to stress, while this change was perhaps due to rearrangement of the cytoskeleton in the embedded osteocyte [21]. E11/gp38, a protein expressed during the rearrangement of the cytoskeleton, such in osteoblast-osteocyte differentiation [80], was enriched in OA, especially in the sclerotic lesions [81]. The elongation of osteocytic dendrites in response to mechanical strain was believed to be regulated by E11 [80]. Changes in the canaliculi then followed, which altered the LCN network. Osteocyte lacunae in OA also appeared small, concentrated, and aligned to the principal stress direction [82]. Osteocytes embedded in large lacunae were thought to have higher mechano-sensitivity, as they responded to mechanical loading with a more prominent drop in sclerostin production and significantly higher β-catenin expression [83]. Decreased DMP-1 expression was also associated with large lacunae and a reduced number of canaliculi [84]. As small lacunae again indicated decreased mechano-sensitivity, local bone remodeling in load-bearing regions might favor osteocyte lacunae with lower mechano-sensitivity in adaptation to repetitive or excessive loading [85]. 

Like the trabecular bones, osteocytes are also well-aligned in OA, likely in response to the cumulative mechanical loading. Osteocyte lacunae align themselves with collagen fibers, which are produced by the ancestral osteoblast [13]. Contractions of actin filaments of the cytoskeleton are believed to introduce intracellular tension in osteocytes. This allows the cell to sense the stress exerted by the extracellular collagen matrix [86]. Collagen orientation is in turn controlled by the dominant loading orientation, either longitudinally with tensile stress or transversely with compression [87,88,89]. Degradation of unloaded collagen fibers by MMPs and preservation of loaded fibers are proposed to mediate collagen alignment (Figure 5b) [90,91]. Meanwhile, fibronectin and integrins are involved in collagen synthesis along with other molecules, which may suggest interactions between cells and ECM to regulate alignment [92]. Sclerostin inhibition was also correlated with high collagen alignment [93], consistent with findings suggesting that periostin was responsible for aligning osteocyte lacunae. However, periostin had no significant effect on lacunae volume [94], which could suggest that it was not the cause of sclerostin-induced modifications of the LCN network. It should also be noted that in contrast to the highly aligned osteocyte lacunae and low BMD observed in end-stage OA, collagen fibers were disorganized, while the subchondral bone was hyper-mineralized in early-stage OA [43]. This could indicate that mineral deposition, rather than the realignment of osteocytes, was the initial response to excessive loading and that there existed dissimilar responses to mechanical loading throughout OA progression.

Another cause of sclerostin deficiency and well-aligned osteocyte lacunae may be osteocyte death. Osteocyte lacunae respond to load from perpendicular and parallel directions differently. In the case of tensile load, osteocytes are more susceptible to microdamage when aligned perpendicularly to the loading direction, while the opposite is true for compression [88,89]. Thus, the pre-existing osteocytes not aligned properly with the principal stress direction are subject to microfracture. Microfractures are believed to induce ionic flux that can activate osteocytes [13] or cause osteocyte apoptosis in the damaged region [17]. Selective osteocyte apoptosis may hence indicate that bone remodeling in accordance with the loading orientation will be preferred. Induced apoptotic osteocytes have shown abnormal gene expression resembling that seen in OA [95], which supports the role of osteocyte apoptosis in overload-induced OA progression. Therefore, in addition to stress-induced sclerostin inhibition, osteocyte apoptosis may also contribute to sclerostin deficiency, which in turn affects osteoblast differentiation and consequently collagen production [96,97]. Recent studies have also revealed similarities between OA and osteonecrosis, especially in the spatial distribution of osteocytes [98]. Increased osteocyte apoptosis was detected with TUNEL staining [28], while the proportion of osteocytes stained positive for activated Caspase-3 was similar to that of normal bones [99]. This might suggest that necrosis, in addition to apoptosis, was responsible for osteocyte death in OA [100]. Past findings also supported that microfracture could induce osteocyte necrosis [101] and contribute to bone remodeling. Upon damage, osteocytes might release damage-associated molecular patterns (DAMPs), which then activated osteoclasts through Mincle, a DAMP sensor. In this way, osteoclasts could sense osteocyte necrosis and initiate bone resorption independent of the sclerostin-RANKL pathway [102]. Hence, osteocyte death induced by overload or microfracture might explain the thinning of the subchondral bone in early-stage OA and the uncoupled osteoblastic and osteoclastic activities [8].

### 3.2. Regulation of Osteoblasts

One of the major functions of osteocytes is acting as the key regulator of osteoblast differentiation. Through the secretion of sclerostin, osteocytes modulate osteoblastic activities and maintain bone homeostasis and integrity [96,97]. Sclerostin can inhibit LRP5/6, the activators of Wnt/β-catenin signaling for mechano-transduction (Figure 1c) [103,104]. The percentage of sclerostin-positive osteocytes is significantly lower in OA [105], whereas the fraction of osteocalcin-positive cells is higher, indicating increased osteoblastic activities [106]. While the number of osteocyte lacunae increases upon mechanical loading, the small individual lacuna size and the low level of sclerostin secretion both indicate the status of immature osteocytes [13,99]. Hence, decreased sclerostin expression in early-stage OA may arise from high mechano-sensitivity of osteocytes and osteocyte death, while the erroneous regulation of osteoblast differentiation may contribute to the low level of sclerostin at end-stage OA. Sclerostin deficiency is believed to activate Wnt signaling pathways and thus increase subchondral bone formation, eventually aggravating OA [13,25]. Mature osteocytes secret sclerostin to negatively regulate the differentiation of osteoblasts lineage and then terminate the cycle of bone remodeling [17,22,23,24]. Immature osteocytes with reduced sclerostin expression in OA would inhibit the negative feedback to MSCs, which were recruited and committed to osteoblast differentiation and bone formation [107]. This was supported by the significantly large number of osteoblasts and osteocytes in osteoarthritic subchondral bone, and by the presence of Runx-2-positive or osterix-positive bone cell clusters in marrow space or adjacent to trabeculae. Osteocalcin, Runx-2, or osterix-positive cells from the subchondral bone were also seen to cluster in the load-bearing regions of OA joints. This suggested upregulated activities of both osteoblasts and pre-osteoblasts [108], possibly in response to the increased demand for bone formation due to excessive mechanical loading and osteocyte death. Igf1 is believed to be involved in osteoblast differentiation mediated by miR-29b-3p (Figure 4). Its expression could be reduced by miR-29b-3p, whose production by osteocytes is downregulated upon mechanical loading. Therefore, increased osteoblast differentiation is closely linked with altered secretion of signaling molecules by osteocytes [109].

Osteoblasts and the matrix they deposit can determine the initial size and shape of the descendant osteocyte lacunae [110,111]. Notably, osteocytes derived from dynamic osteogenesis (DO), where migrating osteoblasts differentiate into osteocytes in apposition to pre-existing trabeculae, are smaller and ellipsoidal [13,112]. As DO occurs commonly in response to mechanical deformation and produces osteocytes aligned with the principal loading direction [113], one may assume that the direct recruitment of osteoblasts, in addition to mesenchymal progenitors, is responsible for subchondral bone remodeling in OA. Hence, the abundance of immature osteocytes in OA can also arise from a problematic transformation from the recruited osteoblasts to osteocytes [71]. Cytoskeleton and the distribution of actin filaments and microtubules are believed to affect the differentiation from osteoblasts to osteocytes [114]. Cytoskeletal rearrangement normally allows osteocytes to adopt different shapes and become more sensitive to mechanical stress and strain. Compared with osteoblasts, osteocytes exhibit decreased stiffness and are more dependent on actin filaments instead of microtubules, though both are critical for mechano-transduction [21,75]. Increased expressions of cytoskeletal proteins in osteocytes were induced by shear stress, along with elevated integrin, E11/gp38, and Runx-2 levels [115]. Hence, cytoskeletal arrangement and thus osteoblast differentiation may be disrupted by an abnormal level of stress in OA (Figure 6a). Meanwhile, osteocyte dysfunction induced by advanced glycation end products (AGEs) showed increased production of sclerostin and RANKL [95]. This process was mediated by FOXO1, a regulator of osteoblast differentiation. FOXO1 can enhance Runx-2 activities [116], while Runx-2 can upregulate sclerostin level [117], inhibit osteoblast-osteocyte differentiation, but promote osteoblast differentiation [118]. Hence, osteocyte dysfunction is linked with erroneous osteoblast differentiation in both mechanically and chemically induced OA. DMP-1 is also considered essential in both osteoblast differentiation and the transformation from osteoblasts to osteocytes [84]. Its release by osteoblasts into the ECM is critical to bone mineralization and osteoblast differentiation [51]. Hence, DMP-1 deficiency in osteoblasts upon mechanical loading [53] may explain the arrested differentiation, though it remains unelucidated how the osteocyte-derived DMP-1 may affect this process. 

In addition to aberrant protein and RNA expressions, anomalies of the content in the ECM may be equally crucial to the pathophysiology of OA. For instance, morphological changes of bone mineral particles and disorganization of collagen fibers are likely involved in osteoblast and osteocyte dysfunction [119]. Hydroxyapatite (HA) is the major type of bone mineral. HA crystals are formed by phosphate and calcium ions, synthesized by osteoblasts and chondrocytes, and regulated by osteocalcin [120]. Osteocytes are also believed to be responsible for bone mineral control through periosteocytic osteolysis, which can be more prominent in pathological conditions [111]. Osteocyte-derived sclerostin can modulate osteoblast-osteocyte differentiation through enhancing phosphorylated MEPE or suppressing phosphate-regulating neutral endopeptidase (PHEX) expression. Cleaved phosphorylated MEPE can bind to HA crystals and halt mineralization, while PHEX can inhibit this process and promote mineralization [121]. Hence, sclerostin deficiency can also affect HA crystals, leading to increased mineralization. Meanwhile, decreased DMP-1 expression associated with Wnt signaling will likely impact the formation of HA crystals, as DMP-1 is essential for the mineralization of collagen [52]. Examinations of HA crystals by our team suggested that bone mineral particles could directly contribute to OA development. In OA bone, HA crystals were predominantly in a cigarette-like shape, while those extracted from the control group appeared bean-like (Figure 7). Abnormal collagen and mineral production might reflect osteoblastic dysfunction, indicate erroneous osteoblast differentiation, and directly or indirectly correlate with osteocyte dysfunction [6,21,80]. When mixed with MG-63 mature osteoblasts, HA nanoparticles derived from OA bone showed a significant inhibitory effect on proliferation [122] and alkaline phosphatase (ALP) activities. Therefore, malformed HA nanoparticles in OA bone might further impair mineral deposition and disrupt matrix maturation by interfering with osteoblastic and pre-osteoblastic activities [123]. Aberrant mineralization could thus exacerbate osteocyte dysfunction and eventually result in the undermined strength of OA trabecular bone.

### 3.3. Regulation of Mesenchymal Progenitors

The recruitment of mesenchymal progenitor cells in joint tissues has long been believed to be involved in OA progression. It was reported that mesenchymal progenitor cells were present at the destruction site of the osteoarthritic cartilage and populated in the superficial layer of the articular cartilage [124,125]. They could also be isolated from the synovium and synovial fluids in OA joints [126,127]. Osteocytes regulate bone-marrow-derived cells through sclerostin, as altered sclerostin expression was found to affect adipogenesis and osteogenesis [79]. TGF-β signaling is associated with the increased number of MSCs in the bone marrow. Enhanced osterix expression in induced OA is related to the phosphorylation of Smad2/3 through TGF-β signaling (Figure 6b) [45]. Osterix normally regulates osteoblast differentiation and promotes sclerostin production [117]. However, increased osterix expression, but not sclerostin expression, has been observed in sclerotic bones. This might be explained by the suppression of TGF-β signaling upon mechanical loading [128]. Suppressed TGF-β signaling could have inhibited the phosphorylation of Smad2/3, decreased sclerostin deficiency, and activated Wnt signaling [129]. MSCs recruitment and osteoblast differentiation were hence enhanced, leading to increased osterix expression by osteoblasts and MSCs, but could not rescue sclerostin deficiency. Osteoblastic differentiation of MSCs is also mediated by Igf1 through the activation of the PI3K/AKT/mTOR pathway [130]. Increased Igf1 production in induced OA models [36] might induce osteogenic differentiation of MSCs and eventually facilitate the maturation of osteoblasts. Osteocytes at the bone formation sites expressed immature markers like E11 in OA samples [81], indicating incomplete differentiation. These findings further suggested that MSCs acted as a reservoir of osteoblasts and were regulated by the newly synthesized immature osteocytes. Hence, microfracture of the subchondral bone by mechanical loading could induce osteocyte death, causing compromised sclerostin expression. Abnormal production of signaling molecules then activated and recruited marrow or periosteum-derived progenitor cells to the boundary of subchondral bone and articular cartilage for tissue repair [131]. 

It has been well documented that the content of the ECM can either induce or inhibit bone formation. For example, the size of bone matrix particles and the structure of collagen fibrils were found to affect the lineage commitment of MSCs [132]. Meanwhile, the shape, crystallinity, surface charge, solubility, and mechanical properties of HA nanoparticles could also influence osteoblastic activities and the osteogenic potential of MSCs [133,134,135,136,137]. It was theorized that MSCs were recruited to osteoclast-mediated bone resorption pits and contacted with the debris after resorption. HA nanoparticles were postulated to be one of the major components of the debris. Differentially shaped HA nanoparticles were suggested to provide dissimilar geometric cues and determine the fate of osteoblasts and MSCs. This was supported by findings suggesting that cigarette-like HA nanoparticles in OA bones could promote cytoskeletal changes of human MSCs and enhance their osteogenic potentials [122]. As there were no significant differences between the HA nanoparticles in healthy and OA bone except for their shape, it is possible that the physical property of HA, rather than the chemical property, is deterministic for controlling the fate of MSCs in bone remodeling. Hence, osteocyte dysfunction may also indirectly promote osteogenic differentiation of MSCs through inducing aberrant mineralization by osteoblasts. 

In summary, altered sclerostin expression may cause osteoblast dysfunction and consequently abnormal mineral production, which can directly or indirectly facilitate osteogenic differentiation of MSCs. It was thus conceptualized that the accumulated microdamage to the subchondral bone by mechanical overload would promote tissue repair, and eventually, bone sclerosis, by which bone turnover was enhanced and resulted in disease deterioration [138]. The increased MSCs recruitment and osteoblastic activities appear to suggest an adaptation to compensate for osteocyte dysfunction and death upon mechanical loading, though this may, in fact, exacerbate the anomalies in tissue morphology and protein production as mature and functional osteocytes are depleted.

### 3.4. Regulation of Cartilage Tissues

Recently, chondrocytes were proposed to be another source of osteoblasts in addition to MSCs, especially since chondrocytes in OA and MSCs appeared to share the fate of hypertrophic differentiation followed by ossification [139,140,141]. Runx-2 and osterix, the regulators of osteoblast differentiation, were assumed to be involved in the possible trans-differentiation from chondrocytes to osteoblasts. Runx-2 could induce chondrocyte hypertrophy, while osterix was expressed by hypertrophic chondrocytes [118,140]. Hypertrophic chondrocytes also express DMP-1, suggesting commitments to the osteogenic lineage and possibly the status of pre-osteoblasts [9,142]. Wnt/β-catenin signaling enhances Runx-2 production and promotes the expression of type X collagen, a hypertrophic marker [143]. Activation of Wnt signaling can be regulated by either TGF-β, BMP-2, with BMP-2 promoting and TGF-β inhibiting chondrocyte hypertrophy (Figure 6a). Previous studies reported that sclerostin was found in chondrocytes of calcified cartilage tissues in normal joints and non-calcified cartilage in OA [25,144]. Sclerostin was hence speculated to be secreted either by chondrocytes themselves or by osteocytes in the subchondral bone and then entered cartilage tissues [145]. Therefore, osteocytes might also function as a regulator of chondrocyte hypertrophy and even trans-differentiation, which could be another possible reason why cell cluster was found at deep and calcified cartilage [28]. It is believed that sclerostin could also inhibit the Wnt/β-catenin signaling pathway and thus downregulate Runx-2, type X collagen, and MMP expressions in cartilage tissue [144,146]. Thus, decreased sclerostin expression might contribute to increased chondrocyte hypertrophy and cartilage degeneration in OA. Notably, sclerostin expression has been observed in cartilage to peak at early-stage chondrogenic differentiation but was decreased at late-stage chondrogenic differentiation [144]. This was echoed by the time-course variation of sclerostin level in both calcified cartilage and subchondral bone seen in induced murine OA models [25]. Differential regulations of sclerostin expressions might explain the increase in chondrocyte sclerostin level observed in mid-stage but not end-stage human OA and could suggest that osteocyte dysfunction preceded or occurred simultaneously with chondrocyte hypertrophy, causing increasingly severe disturbances to the subchondral bone and cartilage as OA progressed [145]. Hence, the disrupted negative feedback to chondrocytes due to a reduction in sclerostin level may contribute to or at least aggravate chondrocyte hypertrophy, though whether trans-differentiation from chondrocytes to osteoblasts and OA progression are mediated by osteocytes and sclerostin requires further investigation.

### 3.5. Regulation of Osteoclasts

Osterix-positive pre-osteoblasts are often spatially disassociated with TRAP-positive osteoclasts in OA bone remodeling, which might explain why bone formation in OA is not coupled with bone resorption pits [28]. Upregulation of osteoclastic activities in OA is believed to be controlled by osteocytes through sclerostin and the OPG/RANKL/RANK system (Figure 8) [13]. This was supported by the elevation in RANKL expression in induced murine OA models [95]. Osteocytes are the major source of RANKL in bone and are responsible for osteoclastogenesis and bone resorption [147]. Osteocyte-derived sclerostin can promote osteoclastogenesis, which is enhanced by RANKL and suppressed by OPG [148]. The PHEX/MEPE/ASARM-PO4 pathway partially regulated by sclerostin was also found to modulate RANKL/OPG expressions. ASARM-PO4, a phosphorylated peptide derived from MEPE, could inhibit RANKL expression but upregulate OPG, while PHEX mostly promoted RANKL expression [149]. Hence, sclerostin deficiency might cause reduced ASARM-PO4 production and increased PHEX signaling [121], thus upregulating the RANKL/OPG ratio. However, a high RANKL/OPG ratio did not guarantee osteoclastogenesis, which might suggest that PHEX/MEPE/ASARM-PO4 signaling could regulate osteoclastic activities through other pathways. For pathological conditions associated with aberrant bone mineralization and protein production, such as hypophosphatemic rickets, the addition of exogenous PHEX might also contribute to higher OPG levels [149]. This might suggest that osteoblastic and osteoclastic activities were also affected bone mineralization. Inflammatory factors commonly found in arthritis could also increase RANKL and decrease OPG expressions, thus upregulating the RANKL/OPG ratio. This effect could be attenuated by mechanical loading [150], which might suggest that osteoclastic activities were favored by inflammation but suppressed by mechanical stress. Similarly, mechanical loading itself could cause a reduction in osteoclastogenesis through increasing MEPE expression [151]. Interestingly, both RANKL and OPG transcriptions were upregulated by mechanical loading, though the RANKL/OPG ratio was decreased. The release of OPG by osteocytes was upregulated soon after mechanical loading, while the release of RANKL was delayed [152], which could suggest a return to homeostasis or regulation by other pathways. Fatigue loading associated with microdamage and increased osteocyte apoptosis, however, appeared to promote osteoclastogenesis, especially near the site of damage [153]. Hence, damage and osteocyte death are likely critical to induce osteoclastogenesis. In the absence of cell and tissue injury, mechanical loading may preferentially enhance osteoblastic activities.

Past studies reported that MSCs and immature osteocytes also expressed OPG, the RANKL inhibitor, while RANKL expression was not affected [81]. As RANKL expression is strongly correlated with osteocyte death, one may assume that osteocyte dysfunction or apoptosis is required to trigger RANKL expression [154], while OPG expression is predominant in osteoblasts and early osteocytes. Thus, temporally and spatially differential regulation of osteoclastogenesis can be achieved depending on the status of osteocytes [17]. Meanwhile, osteoclastic activities can also be mediated by RANKL-independent pathways, such as by DAMPs released during osteocyte necrosis [102], which promotes osteoclastic activities; or by cysteine-rich protein 61 (CYR-61), which inhibits osteoclastogenesis [155]. Localization of living osteocytes in OA suggested that cells were absent from the cores of the enlarged trabeculae but present in the newly formed bone structure in apposition to pre-existing dead trabeculae [98]. Thus, osteocyte death in trabecular cores and osteocyte activities in the surroundings might be correlated with the disassociation between osteoclastic and osteoblastic activities. Likewise, RANKL expressions were generally closer to the site of damage and apoptotic osteocytes, while OPG-positive cells were farther away [153]. These might also partially explain our findings regarding bone remodeling in hip OA. Osteoid formation in apposition to pre-existing trabeculae was not accompanied by bone resorption, while osteoid formation in the control group was closely associated with bone resorption pits spatially (Figure 3c). Meanwhile, in induced murine OA models, bone loss was more prominent at early-stage OA, while bone formation became significant at end-stage OA [156]. Therefore, there appeared to be both spatial and temporal differences in osteoclastic and osteoblastic activities, with osteoclastic activities seemingly preceding osteoblastic activities and showing different site preferences. Hence, OA development may be the result of temporally and spatially differential regulations of osteoblastic and osteoclastic activities by osteocytes in response to mechanical loading and other stimuli. 

## 4. Allometry

Allometry, or biological scaling, refers to the process where one organismal trait varies with another, typically with an emphasis on the correlation between cell size and mass, shape, growth rate, or metabolic rate [157]. In most mammalian cells, cell size and growth rates are believed to be controlled by metabolism, where mitochondrial activities are postulated as the potential sensing mechanism of cell size. Increased cell size is often accompanied by compromised mitochondrial metabolism and upregulation of cytoskeletal genes [158,159], while the optimal cell size is normally of intermediate volume and yields maximal functionality [160]. Chondrocyte hypertrophy, for instance, is known to be associated with OA development. Hypertrophic chondrocytes secreted MMP-13 but expressed fewer cartilage makers, which was linked with increased catabolic activities and reduced anabolic signaling [63,161]. As free energy was required for protein synthesis, a preference for catabolism over anabolism might imply compromised metabolic activities, which was consistent with the observed metabolic inefficiency in other oversized cells [159,160]. Hence, changes in the protein expression profile of an enlarged chondrocyte might correlate with its inability to maintain the metabolic efficiency of a healthy cell. It should also be noted that the variation in metabolism and thus protein synthesis could be phase-dependent, as three distinct phases were previously identified in chondrocyte hypertrophy. Only the second phase was characterized by a drop in dry mass density and presumably low metabolic rates [58]. Meanwhile, the third phase of chondrocyte enlargement was found to be regulated by Igf1 (Figure 4), which was usually secreted by hepatic cells and bone cells, including osteocytes. Upregulation of Igf1 production by osteocytes was observed in response to mechanical loading, especially in the loaded region [58,59]. As Igf1 expression by osteoarthritic chondrocytes themselves were reduced [61], this might suggest that chondrocyte hypertrophy was regulated through osteocyte-derived Igf1 and that Igf1 overexpression by osteocytes could link cartilage degeneration by MMP-13 with excessive loading.

In addition to chondrocytes, osteocytes could also change the size of their lacunae even after being embedded into the bone matrix. This might at least partially explain the morphological and functional changes in osteocytes in response to stimuli [88]. For example, in avian bones, mechanical loading appeared to induce osteocytes responsible for sensing compressive stress to grow along the longitudinal axis, though this effect was modest and did not affect the overall lacuna size. The volume of osteocytes was found to scale with mass-specific basal metabolic rate, though their correlation was relatively weak [110]. An abnormally small lacuna size and an increased number of osteocytes were observed both under excessive mechanical loading and in OA bones [21]. The altered protein expression patterns, such as sclerostin deficiency, might be associated with aberrant metabolism, which then disrupted the regulation of bone cells and joint homeostasis. Meanwhile, the ellipsoidal osteocytes predominantly produced under mechanical loading might suggest elevated metabolism. Elongated cells are often linked with a higher surface-to-volume ratio and metabolic rate compared with spherical cells of the same volume, as the intracellular distances traveled by biomolecules are reduced [160]. Hence, future studies should investigate whether there exists a link between signaling molecule production, cellular metabolic rate, and cell size and whether mechanical loading can alter osteocyte function by changing osteocyte morphology and impacting cell metabolism. 

Overall, in addition to increased osteocyte apoptosis upon damage and osteoblast differentiation for bone repair, mechanical loading may also contribute to osteocyte dysfunction, chondrocyte hypertrophy, and thus OA development through acting on cell volume and metabolism directly or indirectly. Future research should also elucidate whether osteocytes display abnormal metabolic rates under mechanical loading and whether the observed morphological and functional changes can contribute to OA before or independent of osteocyte death.

## 5. Conclusions

In a healthy individual, bone remodeling occurs continuously and actively to maintain homeostasis, notably of the articular joint [17]. This process is believed to be regulated by osteocytes, which recruit and activate osteoblasts, osteoclasts, and progenitors with an osteogenic potential to designated sites for bone formation or absorption. Sclerostin expressed by osteocytes, bone minerals deposited by osteoblasts, and MMPs are among the molecules proposed to affect cellular activities and joint integrity [96,97]. A balance of osteoblastic and osteoclastic activities ensures the mechanical strength and uniform distribution of trabeculae at the tissue level, which enables the joint to withstand compressive forces. If disrupted, bone pathologies such as OA or OP may manifest. The pathomechanism of OA is involuted and likely involves exogenous and indigenous factors, such as trauma and age-related oxidative stress [63,162]. In this review, we attempt to provide a possible mechanism of OA development, with an emphasis on the role of mechanical loading in pathogenesis. At the molecular level, excessive loading can affect protein synthesis, notably inhibiting sclerostin expression, which may partially explain the observed drop in sclerostin level in OA [71]. This process interacts with various signaling pathways, possibly including periostin, DMP-1, and TGF-β signaling, and may be associated with aberrant metabolic activities [29,74,128]. Meanwhile, the low level of sclerostin expression may also arise from the deaths of pre-existing osteocytes due to microdamage and their replacement by the newly synthesized immature or dysfunctional osteocytes. Sclerostin deficiency then activates the Wnt signaling pathways and regulates the OPG/RANKL/RANK system, which recruits osteoclasts and osteoblasts in a time-and-space-dependent manner [13]. It is hypothesized that osteoclasts will first absorb the existing bones in early-stage OA, while osteoblasts will deposit collagens and minerals to align the osteocytes along the principal loading direction and enhance the osteogenic potential of mesenchymal progenitors in end-stage OA [98]. Thus, the spatially and temporally uncoupled bone remodeling renders the trabecular bone mechanically inferior and results in structural disturbances of the subchondral bone. Abnormal bone microarchitecture may undermine the ability of the joint to sustain compression and gradually contribute to cartilage degeneration.

In summary, as mechano-sensors, osteocytes display aberrant and altered protein secretion under excessive mechanical loading, which is believed to link osteocytes with the pathophysiology of arthritis, including ankylosing spondylitis and osteoarthritis [82,163]. Thus, osteocyte dysfunction and the relevant signaling pathways may serve as a potential therapeutic target for OA treatment, especially if induced by cumulative or repetitive loading [18]. However, whether osteocytes and their secretions remain to be the central regulator and an essential signaling molecule in other means of pathogeneses of OA is not fully elucidated and should be further investigated in the future. 

## Figures and Tables

**Figure 1 ijms-22-06522-f001:**
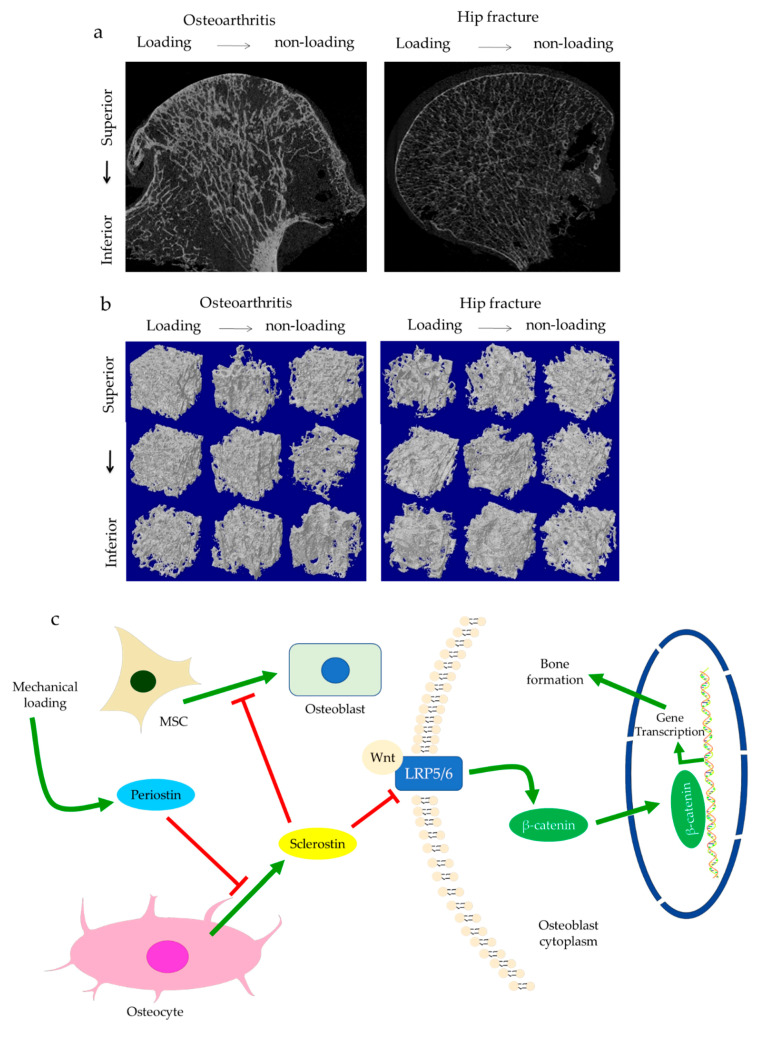
The representative uCT images and data show the comparisons of hip osteoarthritis (OA) with a fracture in two- (**a**) and three-dimensional architecture (**b**). Osteoarthritic bone was characterized by high bone mass disproportionately concentrated in the load-bearing regions in comparison with hip fracture bone. This is believed to be regulated by periostin and sclerostin (**c**). Enhanced periostin and decreased sclerostin expressions upon mechanical loading can activate Wnt/β-catenin signaling through LRP5/6. β-catenin then promotes gene transcription in the targeted cell, such as osteoblasts, which results in bone formation.

**Figure 2 ijms-22-06522-f002:**
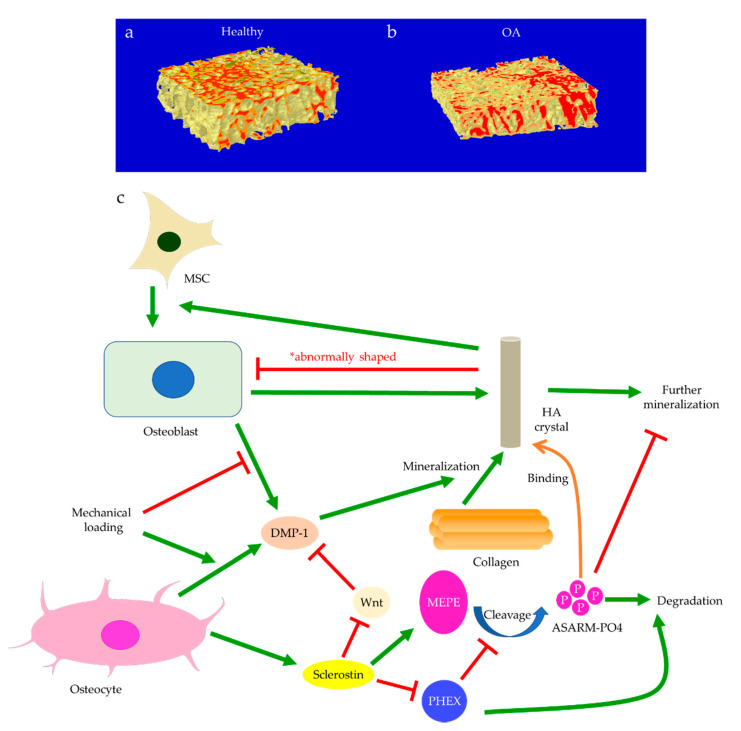
The representative micro-CT images show the microstructure of trabecular bone in the medial plateau of healthy (**a**) and osteoarthritic (OA) knees (**b**). The color-coded images of trabecular bone show the distribution of bone mineral density (BMD). The red-labeled trabecular bone represented high BMD. Bone mineralization is regulated by sclerostin, dentin matrix protein 1 (DMP-1), matrix extracellular phosphoglycoprotein (MEPE), and phosphate-regulating neutral endopeptidase (PHEX) (**c**). Osteoblast-derived DMP-1 is critical for the mineralization of collagen fibers, while sclerostin can regulate MEPE and PHEX to control further mineralization of hydroxyapatite (HA) crystals. Mechanical loading regulates DMP-1 expressions by osteoblasts and osteocytes differently. Abnormally shaped HA crystals can induce osteogenic differentiation of mesenchymal stem cells (MSCs) or trigger osteoblast dysfunction.

**Figure 3 ijms-22-06522-f003:**
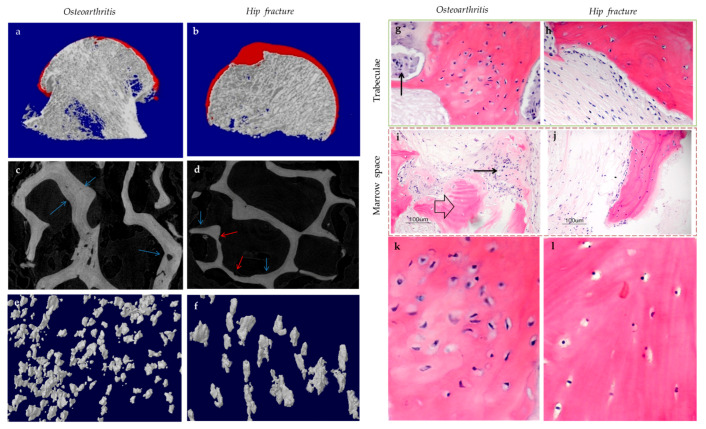
The representative uCT (**a**–**f**) and histological images (**g**–**l**) show the comparisons between hip osteoarthritis (OA) (**a**,**c**,**e**,**g**,**i**,**k**) and fracture (**b**,**d**,**f**,**h**,**j**,**l**). As shown by uCT images, osteoid formation (blue arrow) was found in apposition to pre-existing trabeculae for both hip OA (**c**) and fracture (**d**). The newly formed bone was associated spatially with bone resorption pits (red arrow) in hip fracture, yet it was not the case for hip OA. It was revealed by histology that bone cell clusters (black arrow) were present close to osteoid formation (block arrow) either in trabecular bone (**g**) or marrow space (**i**). In contrast, bone cell clusters were not found in hip fractures (**h**,**j**). The two (**k**,**l**) and three (**e**,**f**) dimensional morphology of osteocytes lacunae both revealed that the number of lacunae from hip OA bone was higher than that of hip fracture. In addition, OA osteocytes lacunae were well aligned and more plate-like as compared with hip fracture ones. (**g**–**j**: Hematoxylin and eosin staining with a magnification of 20×).

**Figure 4 ijms-22-06522-f004:**
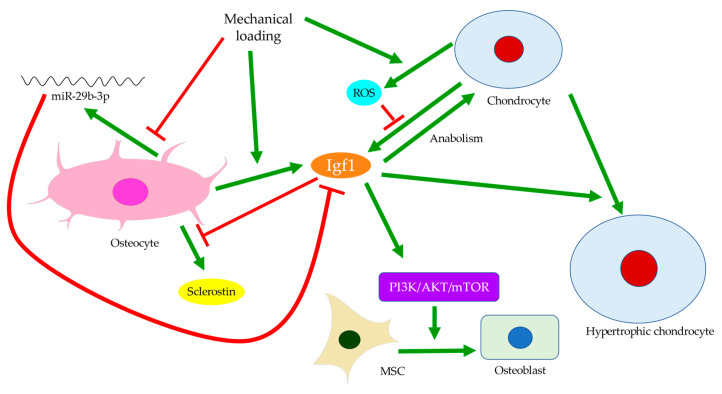
Insulin-like growth factor 1 (Igf1) signaling in the crosstalk between osteocytes and chondrocytes. Mechanical loading regulates Igf1 expressions by osteocytes and chondrocytes differently. Inhibition of miR-29b-3p production by osteocytes can increase Igf1 expression. Meanwhile, elevated reactive oxygen species (ROS) production by chondrocytes can suppress Igf1 expression. Igf1 also regulates sclerostin expression, chondrocyte hypertrophy, and osteogenic differentiation of MSCs through the PI3K/AKT/mTOR pathway.

**Figure 5 ijms-22-06522-f005:**
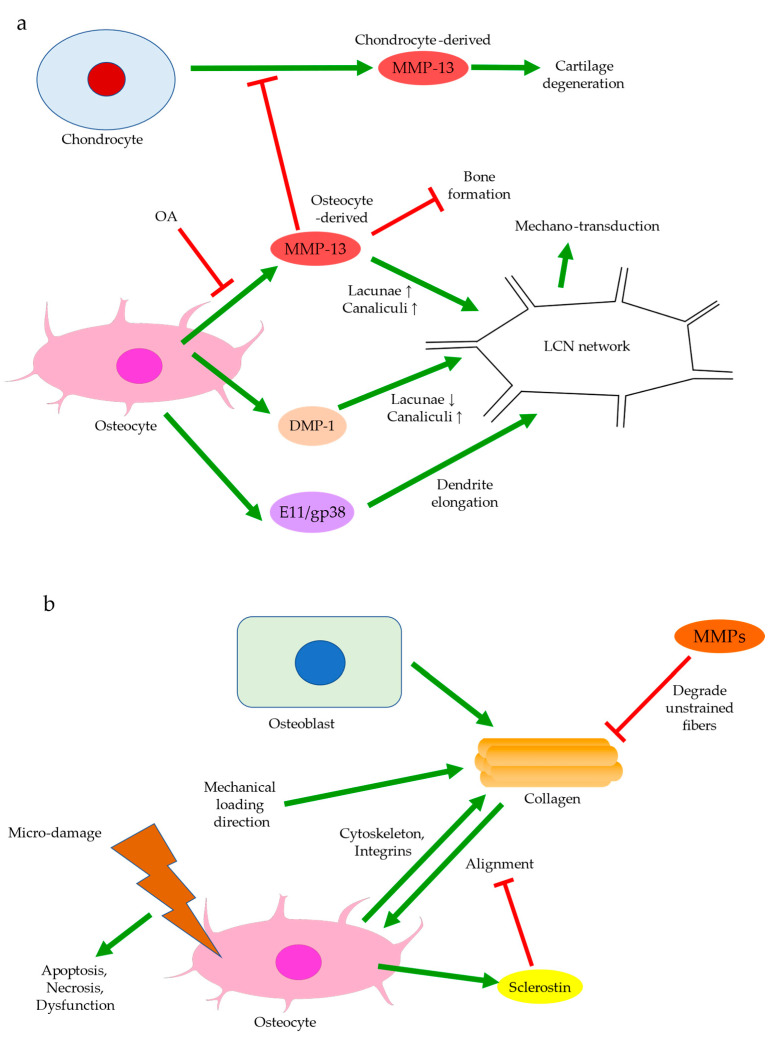
Regulation of osteocyte morphology (**a**) and alignment (**b**). Osteocyte-derived DMP-1 and E11/gp38 can alter the lacuno-canalicular networks (LCN) and hence mechano-transduction. In OA, matrix metalloproteinase (MMP)-13 production by osteocytes is decreased. This may contribute to increased bone formation in subchondral bone and elevated MMP-13 expression by chondrocytes (**a**). Mechanical loading can determine the alignment of collagen fibers, possibly by degrading the unstrained fibers. Collagen alignment, in turn, affects the alignment of osteocyte lacunae through cytoskeleton and integrins. Sclerostin expression is more prominent in unaligned osteocytes. Micro-damage may induce dysfunction, apoptosis, or necrosis of the unaligned osteocytes (**b**).

**Figure 6 ijms-22-06522-f006:**
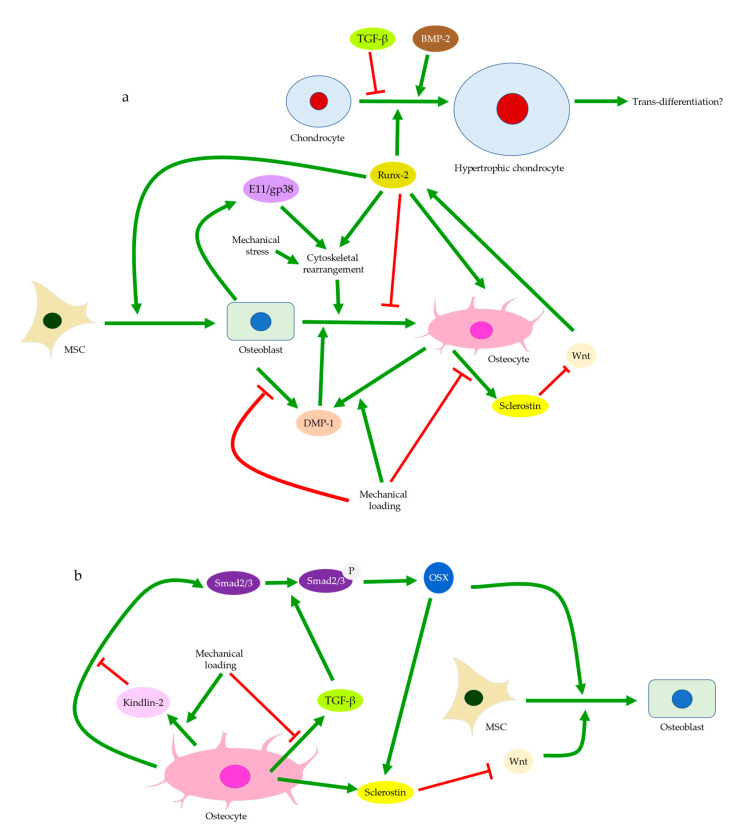
Osteoblast differentiation and osteoblast-osteocyte transformation (**a**) and the regulation by TGF-β signaling (**b**). Mechanical loading can promote osteogenic differentiation of MSCs. This is regulated by sclerostin and Wnt signaling and likely involves Runx-2 expression. Rearrangement of the cytoskeleton and DMP-1 are both believed to regulate osteoblast to osteocyte transformation (**a**). Osteoblast differentiation can also be regulated by TGF-β signaling and phosphorylation of Smad2/3. Mechanical loading can suppress Smad2/3 expression or their phosphorylation, which can lead to sclerostin deficiency and impact osteoblast differentiation (**b**). Chondrocyte hypertrophy is believed to be regulated by factors including Runx-2, BMP-2, TGF-β, and sclerostin, which is possibly enhanced by mechanical loading. Hypertrophic chondrocytes are speculated to possess the potential to trans-differentiate into osteoblasts.

**Figure 7 ijms-22-06522-f007:**
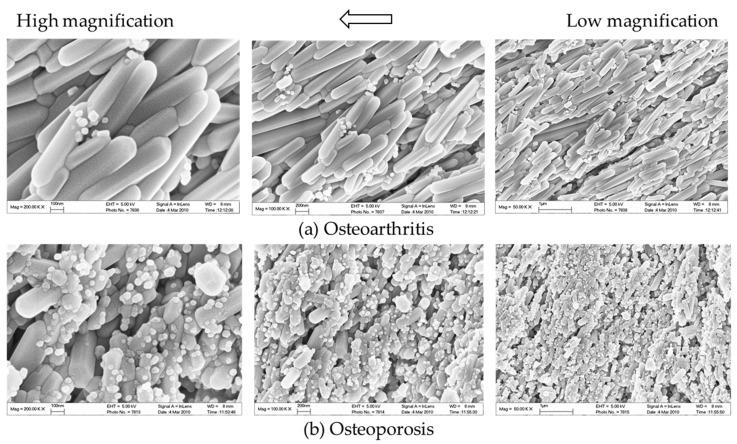
The representative SEM images show the differentially shaped HA nanoparticles of the trabecular bone from donors with (**a**) knee OA or (**b**) osteoporosis at different magnifications. In osteoporosis patients, HA crystals present with a bean-like shape. In contrast, HA nanoparticles from OA patients presented with a cigarette-like shape.

**Figure 8 ijms-22-06522-f008:**
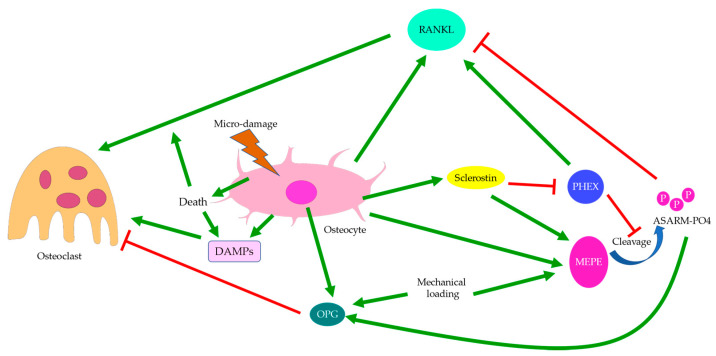
Regulation of osteoclasts. Osteoclastogenesis is regulated through the OPG/RANKL/RANK system and sometimes by damage-associated molecular patterns (DAMPs). The RANKL/OPG ratio can be regulated by PHEX, MEPE, and sclerostin. Sclerostin deficiency may suppress MEPE but enhance PHEX expressions and increase the RANKL/OPG ratio. Osteoclastic activities are associated with mechanical loading and osteocyte death, with osteoclasts more likely to be recruited to the site of micro-damage.

## Data Availability

No new data were generated in this study. Data sharing is not applicable to this article.

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
