# Peer review of "Osteocyte Dysfunction in Joint Homeostasis and Osteoarthritis"

_ijms, 2021, doi:10.3390/ijms22126522_

Round 1

Reviewer 1 Report

The authors did summarize current understanding of osteocytes at the crossroad of allometry and mechanobiology to exploit the relationship of osteophyte morphology and function in the context of joint ageing and osteoarthritis. Although this is well-written comprehensive review, there are some minor concerns which can improve the article. 

1. Preparing a figure to summarize the regulation of osteoblasts and osteoclasts (3.2. and 3.4.) can facilitate reading this article. 

2. If the figures are already published, the source should be clarified. 

3. Recheck references. Some of them are not in line with the right style such as 4, 15, 34, 51, 110, and 111. 

Author Response

Response to Reviewer 1 Comments

Minor Issues:

1)            Prepare a figure to summarize the regulation of osteoblasts and osteoclasts (3.2. and 3.4.) can facilitate reading this article. 

Author Response: We agree that adding figures will help with understanding the molecular pathways for regulating osteoblasts and osteoclasts, and thus we have included figures to explain osteoblast differentiation and osteoclastogenesis mediated by osteocytes.  As we have also added discussion on the molecular mechanisms behind the morphological changes, we included figures illustrating the mechanisms behind subchondral bone remodelling, crosstalk between subchondral bone and cartilage, and morphological changes of osteocytes.

Author Actions:

  • Added Figure 1c to explain bone formation mediated by osteocytes
  • Added Figure 2c to explain the mechanism behind bone mineralization
  • Added Figure 4 to explain the crosstalk between subchondral bone and cartilage
  • Added Figure 5 to explain the mechanism determining osteocyte morphology
  • Added Figure 6 to explain the regulation of osteoblast differentiation
  • Added Figure 8 to explain the regulation of osteoclastogenesis

2)            If the figures are already published, the source should be clarified.

Author Response: The figures were not published previously as they are undisclosed data collected by the authors’ team.

3)            Recheck references. Some of them are not in line with the right style such as 4, 15, 34, 51, 110, and 111.

Author Response: We apologize for the mistake and have modified the reference style accordingly.

Reviewer 2 Report

The article "Osteocyte Allometry in Joint Homeostasis and Osteoarthritis" is made on a relevant topic.

Since the manuscript was submitted to the "International Journal of Molecular Sciences", I expected to see a review of the literature on the molecular basis of the pathogenesis of osteoarthritis. Unfortunately, most of the article describes rather old and well-known results of studying bone tissue by morphological methods. Accordingly, "References" also include a lot of fairly old publications. Only part of the sections "3.2. Regulation of osteoblasts", "3.4. Regulation of cartilage tissues" and "3.5. Regulation of osteoclasts" contain a description of the molecular mechanisms of bone changes. Directly "Allometry" (Title!) is devoted to only 2 paragraphs.

In my opinion, the submitted manuscript cannot be published in the "International Journal of Molecular Sciences". To publish the article, it is necessary to remove the old generally known data and instead present the results of studies that really describe the molecular mechanisms of osteoarthritis and bone variability, show the connection between morphology and molecular biology. Re-review required.

The manuscript also contains minor inaccuracies. For example, "... not Caspase-3 staining ..." (line 193), the authors may have meant staining for detect of Caspase-3.

I also want to see the authors' opinion about the phrase: "The accumulated microdamage to the subchondral bone by mechanical overload would promote tissue repair, and eventually, bone sclerosis, by which bone turnover was enhanced and resulted in disease deterioration."

In addition, I recommend that the authors in the figure signatures also give links to the articles from where these figures were taken for this review.

I am not a native English speaker and it is difficult for me to judge the quality of the manuscript in English. But based on the clarity of the entire text, I can conclude about the acceptable quality of the level of English used.

Author Response

Response to Reviewer 2 Comments

Major Issues:

1)            Most of the article describes rather old and well-known results of studying bone tissue by morphological methods. Accordingly, "References" also include a lot of fairly old publications.

Author Response: We have updated the reference list accordingly.

Author Action: We have removed out-of-date references and included references to new findings in molecular mechanisms

2)            Only part of the sections "3.2. Regulation of osteoblasts", "3.4. Regulation of cartilage tissues" and "3.5. Regulation of osteoclasts" contain a description of the molecular mechanisms of bone changes. Directly "Allometry" (Title!) is devoted to only 2 paragraphs.

Author Response: We agree that this paper discusses extensively on the morphological characteristics of OA without sufficient focus on the molecular basis. We have hence included some of the possible molecular mechanisms behind OA pathogenesis related to mechanical loading.

Author Action:

  • Edited ‘Abstract’ and ‘Introduction’ and included discussion on molecular mechanisms
  • Included the molecular pathways behind ‘Subchondral bone remodelling’ and changes in ‘Osteocyte morphology’
  • Elaborated on the molecular mechanisms behind the ‘Regulation of Osteoblasts’, ‘Regulation of mesenchymal progenitors’, ‘Regulation of chondrocytes’, and ‘Regulation of osteoclasts’ by osteocytes
  • Edited ‘Allometry’ and ‘Conclusions’ in accordance with the above changes
  • Title changed from ‘Osteocyte Allometry in Joint Homeostasis and Osteoarthritis’ to ‘Osteocyte Dysfunction in Joint Homeostasis and Osteoarthritis’ to emphasize on the osteocyte-mediated regulation of OA pathogenesis

3)            To publish the article, it is necessary to remove the old generally known data and instead present the results of studies that really describe the molecular mechanisms of osteoarthritis and bone variability, show the connection between morphology and molecular biology. Re-review required.

Author Response: We agree with the reviewer and have discuss the molecular mechanisms behind OA pathogenesis, including bone formation, mineralization, and cartilage degeneration. (See above)

Minor Issues:

1)            The manuscript also contains minor inaccuracies. For example, "... not Caspase-3 staining ..." (line 193), the authors may have meant staining for detect of Caspase-3.

Author Response: We apologize for the inaccurate description regarding the detection of Caspase-3 activities. We have rephrased the sentence and clarify the finding.

Author Action: Changed “…detected with TUNEL [28] but not Caspase-3 staining…” to “… detected with TUNEL staining [28], while the proportion of osteocytes stained positive for activated Caspase-3 was similar to that of normal bones [99]…” (Line 327-329)

2)            I also want to see the authors' opinion about the phrase: "The accumulated microdamage to the subchondral bone by mechanical overload would promote tissue repair, and eventually, bone sclerosis, by which bone turnover was enhanced and resulted in disease deterioration."

Author Response: We mostly agree with this view and have added our own opinions following the original sentence.

Author Action: Added ‘The increased MSCs recruitment and osteoblastic activities appear to suggest an ad-aptation to compensate for osteocyte dysfunction and death upon mechanical loading, though this may in fact exacerbate the anomalies in tissue morphology and protein production as mature and functional osteocytes are depleted.’ (Line 482-485)

3)            In addition, I recommend that the authors in the figure signatures also give links to the articles from where these figures were taken for this review.

Author Response: As mentioned above, the figures were not taken from published articles as they are undisclosed data collected by the authors’ team.

Round 2

Reviewer 2 Report

I am quite pleased with the responses of the authors of the manuscript to my comments and with almost all the changes made. The article has been completely revised and, in my opinion, has become much better. But...

In the abstract, when proteins, glycoproteins, etc. are mentioned 1 time, it is not necessary to give an abbreviation.

The authors write: "Subchondral bone disturbance has been ...". Maybe they meant "structural disturbances of subchondral bone"? It is necessary to find and correct all such minor inaccuracies.

The authors reported that the presented review contains pictures obtained by them. If the manuscript has its own data, it is desirable, but not necessary, to indicate "the results of the authors' research", "the figure obtained by the authors' team", "However, our data contradict the results ..." or "Our data confirm the results ..."

Unfortunately, when correcting the manuscript, errors were made in the use of English (for example, I do not like "This was thought to precede or occur simultaneously with cartilage degeneration in the spontaneous OA model of guinea pigs"). Translation editing required.
